# Obstructive Sleep Apnoea in Patients with Treacher Collins Syndrome—A Narrative Review

**DOI:** 10.3390/jcm14134741

**Published:** 2025-07-04

**Authors:** Anna Paradowska-Stolarz, Katarzyna Sluzalec-Wieckiewicz, Marcin Mikulewicz, Conrad Maslowiec, Katarzyna Kokot, Lucia Miralles-Jorda, Martyna Sobanska, Joanna Laskowska

**Affiliations:** 1Division of Dentofacial Anomalies, Department of Orthodontics and Dentofacial Orthopedics, Wroclaw Medical University, Krakowska 26, 50-425 Wroclaw, Poland; marcin.mikulewicz@umw.edu.pl (M.M.); j.laskowska@umw.edu.pl (J.L.); 2Private Office, Gajowicka 137/3, 53-407 Wroclaw, Poland; karas60@o2.pl; 3University Dental Center, Krakowska 26, 50-425 Wroclaw, Poland; cjmmaxwell07@gmail.com; 4Department of Conseling, Institute of Pedagogy, University of Wrocław, J.W. Dawida 1, 50-527 Wroclaw, Poland; katarzyna.kokot@uwr.edu.pl; 5Department of Dentistry, Catholic University of Valencia, Quevedo Str 2, 46001 Valencia, Spain; lucia.miralles@ucv.es; 6Department of Orthodontics and Dentofacial Orthopedics, Wroclaw Medical University, Krakowska 26, 50-425 Wroclaw, Poland; martyna.sobanska@umw.edu.pl

**Keywords:** Treacher Collins syndrome, obstructive sleep apnoea, airway management, craniofacial anomalies, mandibular distraction osteogenesis

## Abstract

**Background:** Treacher Collins Syndrome (TCS) is a rare, congenital craniofacial syndrome. Its most characteristic feature is mandibular and midface hypoplasia. Due to malformations of the facial skeleton, airway abnormalities can also be observed, predisposing individuals to obstructive sleep apnoea (OSA). OSA in TCS may contribute to significant morbidity, including developmental delays, cardiovascular disorders and reduced quality of life. **Objectives:** This narrative review aims to present the true prevalence of OSA and the treatment options for TCS patients. Additionally, the pathophysiology and diagnostic tools for this condition were briefly outlined. **Methods:** The literature search included publications from PubMed, Scopus, Web of Science and Cochrane Library. The chosen period of time for these publications was 2000–2024. **Results:** The results showed that OSA is a serious problem among TCS patients. Although there is no standardised treatment protocol, the primary methods often include mandibular distraction osteogenesis (MDO) and continuous positive airway pressure (CPAP). Approaches such as hypoglossal nerve stimulation (HNS) need further investigation, especially with longitudinal observations. **Conclusions:** The development of treatment options seems to be promising, suggesting a favourable outlook for standardising the treatment protocols.

## 1. Introduction

The physiological process of sleep is crucial and has a profound impact on one’s well-being. Thus, an inadequate duration and quality of sleep affect the whole body. The correlation between sleep and good health is obvious—a lack of sleep can negatively influence one’s life, leading to severe health issues [1].

The medical field meets extensive problems caused by the increasing prevalence of obstructive sleep apnoea (OSA)—a disorder involving excessive muscle relaxation during sleep. This causes airway collapse, impairs air flow and can potentially result in possibly life-threatening situations. Primarily, OSA is defined as being a muscular problem that is frequently diagnosed among the general population, by general practitioners as well by dentists [2,3,4]. Furthermore, there is a strong connection between OSA and malocclusions, so it is highly probable that congenital syndromes may also be associated with OSA [5]. Due to the fact that sleeping disorders are common, diagnostic tools are being developed to monitor them. One of the most common tools currently used is polysomnography [6].

Treacher Collins Syndrome (TCS) (or Franceschetti Syndrome in other words) is a rare congenital deformity with a strong genetic background—with four genes being responsible (TCOF1, POLR1D, POLR1C and POLR1B). Mutations in the above-mentioned genes lead to defects in neural crest cell migration [7]. Its birth prevalence is estimated at 1:50,000 live births. It is assumed that 40% of cases have a family history [8,9,10].

Its main facial characteristic features include lower eyelid abnormalities; hypoplasia of the bones (mainly malar, but also mandibular); small, underdeveloped ears; and downward-slanting palpebral fissures. In some cases, clefts can be present [11]. Despite the possibility of diagnosing TCS in utero, this congenital defect often remains unknown to mothers, possibly due to limited awareness during pregnancy. Thus, a prenatal diagnosis might not be made [12,13]. This abnormality is associated with the first and second pharyngeal arches, characterised by one-sided underdevelopment of the facial structures, particularly the zygomatic bones, the mandible and the external ear. Hemifacial micrognathia is also observed. Moreover, midface hypoplasia, hearing loss and other facial features can be identified. Due to severe malformation of the craniofacial region, including retrusion and counterclockwise rotation of the mandible, one of the issues that can be encountered is nocturnal apnoea. Thereby, difficulty breathing can induce potentially life-threatening circumstances [7,8,10,11]. Severe orofacial malformations and associated hearing loss might necessitate speech therapy. Hearing loss itself is likely to guarantee challenges in learning how to speak. Additionally, the base of the tongue is tilted backwards, and due to underdevelopment of the phono-articulatory organs, the epiglottis collapses over the larynx. Muscular and musculoskeletal changes may occur midst communication, resulting in difficulties producing the proper sounds and forming speech. This also leads to communicative misunderstanding, as the understanding of others’ speech could also be limited [14,15]. It is a well-known fact that mandibular growth disturbances can influence the whole skeleton and body; therefore, the position of the mandible is also a potential risk factor for temporomandibular disorders [16,17]. Children suffering from breathing disorders also often present poorer cognitive development and significantly poorer growth. This might result in cardiopulmonary complications but also behavioural problems (hyperactivity, reduced attention and somatic complaints) [18].

The potential treatment options for these patients are surgical procedures, involving both the hard and soft tissues [8,9,11]. Airway obstruction in TCS patients consequently yields a predisposition towards recurrent episodes of upper airway collapse during sleep, known as OSA. Disturbances with breathing at night may lead to hypoxia, hypercapnia and sleep fragmentation [19]. This condition ideally should be treated at an early age. The consequence of not treating this condition may result in life-threatening conditions, growth impairments, problems thriving, developmental delays and cognitive impairments [20,21].

The aim of this narrative review is to present the problem of OSA in the context of patients with Treacher Collins Syndrome. The main objective is to highlight the difficulties that doctors deal with when planning the treatment for patients suffering with TCS. As in any congenital disease, the need for a multidisciplinary approach is huge, and a general understanding of the patient’s main problems is crucial in this situation. The prevalence, diagnosis and treatment methods for OSA in TCS patients will be presented, as will the advancements in its treatment. Possible future research directions and approaches will also be presented.

## 2. Materials and Methods

The PubMed, Web of Science, Scopus and Cochrane Library databases were searched by two independent researchers (A.P.S., K.S.W.) in January 2025 in order to obtain the results. The search words used were “OSA” and “Treacher Collins Syndrome”. According to the Cochrane Library database, no systematic review had previously been carried out on this topic. The abstracts of the articles were skimmed to check the validity of the search. Due to the small number of articles in general, all papers were screened for this topic. If no clear scientific background was noticed, the papers were rejected. This research was conducted according to the PRISMA standards.

### 2.1. The Search Strategy

The PubMed, Scopus, Web of Science and Cochrane Library databases were searched by two independent reviewers (A.P.S. and K.S.W.). The keywords and phrases were “OSA” and “Treacher Collins Syndrome” and are presented in Table 1. The chosen papers were published in the English language. The final search was carried out in January 2025, and papers published between 2000 and 2024 were taken into account. All papers, regardless of the age of the participants, were investigated.

### 2.2. The Inclusion and Exclusion Criteria

Only full articles published in English from 2000 to 2024 were taken into account. Systematic reviews, randomised controlled trials, cohort studies and case series were taken into consideration. The authors focused on studies reporting the pathophysiology, diagnosis, prevalence and treatment of OSA in patients with Treacher Collins Syndrome. Papers with an unclear background were not taken into account.

Articles in other languages were excluded from this study. Additionally, conference abstracts and animal studies were excluded. The presence of other congenital syndromes apart from TCS were also eliminated from the review.

### 2.3. Data Extraction and Analysis

Two independent researchers (A.P.S. and K.S.W.) extracted the data for this study, in accordance with the mentioned inclusion and exclusion criteria. They evaluated the study designs, sample sizes, the methods of diagnosis, the treatment interventions and the general outcomes. Papers were divided based on the prevalence of OSA in TCS patients, its diagnosis and the treatment strategies.

## 3. Results

The search criteria, presented in Figure 1, revealed a total number of 15 papers. In the first search, 17 PubMed papers were found, while in the Web of Science (WOS) database, 11 papers were found. Scopus revealed 13 papers. After setting the years 2000–2025 within the searching criteria, there were 16 papers in PubMed, 8 in WOS and 13 in Scopus. After removing duplicates, 16 papers were taken into account; finally, 1 paper was published in the Chinese language and was therefore rejected, giving a total number of 15 papers written in English.

### 3.1. The Prevalence and Pathophysiology of OSA in TCS Patients

According to the presented reports, 70% of TCS patients suffer from OSA. The severity of this condition increases with age and has a strong connection with the severity of mandibular hypoplasia [2], which was also confirmed by Guilleminault et al. [2] in a longitudinal cohort study. They also stated that due to serious airway problems, early surgical intervention should be mandatory.

The main causes of airway obstruction in individuals with Treacher Collins Syndrome are presented in Figure 2. The primary issue is mandibular hypoplasia with so-called glossoptosis, which is posterior displacement of the tongue. Another issue could be midface hypoplasia, with narrowing of the nasopharyngeal airway. Additionally, microstomia with a cleft palate and impaired airway function, as well as hypotonia of the pharyngeal muscles with airway collapse during sleep, is a further contributor [22].

### 3.2. The Diagnosis of OSA in TCS

Polysomnography (PSG) [23] is the gold standard for OSA diagnosis in general patients, as well as in TCS patients. Unfortunately, the need for hospitalisation in order to carry out this examination gives it limited accessibility. For this reason, other screening tools might be helpful. To identify oxygen desaturation patterns, nocturnal pulse-oximetry can be used [19]. A straightforward way to recognise upper airway obstructions is using a cephalometric X-ray (which should be performed prior to treating an orthodontic patient [24,25]). Another, more advanced method of diagnosis is MRI, allowing for an accurate evaluation of the airway anatomy, thus revealing possible obstruction sites [26]. An interesting tool for OSA screening is video sleep studies, with this method being especially useful in small children, including infants, in whom airway abnormalities could be present [2].

A previous systematic review by Akre et al. [22] emphasised the importance of early diagnosis of OSA in children with TCS. These authors recommended applying PSG as soon as possible to prevent future complications.

### 3.3. Treatment Strategies

According to the American Academy of Sleep Science [27], OSA should be treated non-surgically. This, however, does not apply to patients with congenital diseases, as in such cases, the problem is often more complex. Therefore, the treatment methods for OSA among patients with TCS combine surgical and non-surgical approaches. A summary of the surgical and non-surgical strategies for treatment is presented in Table 2.

#### 3.3.1. Continuous Positive Airway Pressure

CPAP (continuous positive airway pressure) is the method of choice for the non-surgical treatment of OSA. It is a form of positive airway pressure ventilation. It can be carried out as the first-line therapy in mild to moderate OSA cases [20,28]. With air being continuously applied to the upper respiratory tract, a pressure greater than atmospheric pressure is formed, preventing the upper airway from collapsing. This method is thought to be effective in treating this condition and therefore can be used with great success in individuals with TCS. Possible problems include a lack of patient compliance in the use of CPAP therapy, thus making this a limiting factor. According to Askland [28], 8% of people quit using it after the first night, while 50% of individuals cease treatment within the first year of therapy. What may be of key interest for this procedure is the pretty low percentage of recovery after this procedure. This may be caused by the severe anatomical abnormalities present in individuals with TCS. This challenging problem may be contributed to by a poor mask fit and discomfort, as well as the age of the individual [29].

#### 3.3.2. Nasal Airway Stents

Nasal airway stents were introduced into the treatment for OSA in the 1970s [30] to help widen the upper airways. Although successful, they can only be used temporarily until surgical correction is complete. Just like with other stents, they can be used in small children, including neonates, especially ones suffering with severe glossoptosis [31].

#### 3.3.3. Mandibular Advancement Devices

Mandibular advancement devices are the standard devices used to relieve the symptoms of OSA and reduce apnoea and hypopnea [32]. Also, radiographs show that using any mandibular advancement tool, including orthodontic appliances, among them functional ones, might reduce the symptoms of night breathing disturbances and consequently lead to widening of the airways [33]. They can be used as an alternative in less severe cases to surgical intervention and mandibular elongation.

#### 3.3.4. Surgical Interventions

In most cases, surgical interventions are the final method of choice in TCS patients suffering from OSA. Mandibular distraction osteogenesis (MDO) can be used in order to lengthen the mandible, usually when a patient is still growing. Thanks to mandible lengthening, the base of the tongue can be advanced, opening the airways. MDO is more desirable in growing patients, whereas, on the other hand, BSSO (bilateral split sagittal osteotomy) is performed in adults [34]. In both cases, MDO and BSSO reduce the apnoea–hypopnea index (AHI), confirming that a more appropriate skeletal ratio does result in an improved airway width [34,35]. These methods are also successful in correcting malocclusion [36]. Studies in other children with microgenia, as one of the features of Pierre Robin sequence, revealed, however, that MDO can lead to overcorrection and consequently a reversed overjet. Furthermore, patients may still present with some mild symptoms of OSA [37]. This can be also caused by velopalatal insufficiency, especially for individuals with accompanying clefts. This anomaly co-exists with TCS in approx. 25% of individuals [8,9]. The scars after surgical palatoplasty may interfere with life and may also require more advanced, reconstructive procedures [8,38].

Another treatment method used in severe life-threatening cases is tracheostomy. Due to its high morbidity and negative impact on speech [22], this is carried out only in severe cases when other methods cannot be applied. Although it improves breathing and reduces the AHI [39], this method finds more use in adults than children.

A novel, promising therapy is hypoglossal nerve stimulation (HNS or HGNS). The nerves’ anatomy allows for easy surgical access and tolerable neurostimulation [40]. This method helps control the airways through neuromuscular media [41]. Complete concentric collapse (CCC) of the velopharyngeal airway is a common reason why patients require a bilateral procedure, which in this case would be HGNS for the treatment of OSA. Originally, the applied methods only stimulated one side of the tongue and did not interfere with the soft palate and the throat. For this reason, the bilateral procedure could be considered. A lack of diagnosis of CCC before the beginning of the treatment may result in poor results. The use of HGNS results in bilateral stimulation of the tongue and therefore may address CCC management better by more effectively opening the airway [42,43,44]. Its success rate is estimated at 80%, showing promising results; however, because this method has just recently been developed, further research is necessary. Patients treated with HNS show improvements in their symptoms. Moreover, the device is well tolerable and much easier to use than PAP therapy [38]. Although this procedure has recently become more likely to be performed, the first successful attempt to incorporate it into the treatment of individuals with TCS was made by Wong et al. in 2023 [45].

### 3.4. Long-Term Outcomes

The reviewed studies show that early surgical intervention performed in order to treat OSA can lead to general improvements and development in patients with TCS. The growth, cognition and daytime functioning of these individuals improve. However, impairments may still remain, and OSA can recur due to craniofacial growth changes and the inability to control them. The initial improvements with the treatment of OSA using surgical maxillofacial advancement procedures do not seem to be confirmed in the long-term outcomes [46,47,48]. Patients frequently require long-term orthodontic treatment and several surgical procedures, including osteotomies at a young age, which may carry multiple risks. Also, the American Academy of Sleep Science [27] advises the use of a non-surgical approach to treating OSA, so avoiding surgery is also in accordance with these recommendations.

## 4. Discussion

According to the American Academy of Sleep Science [27], the treatment method chosen for OSA should be based, if possible, on non-surgical methods. In patients with TCS, this kind of approach, although theoretically possible, would not be entirely successful. The aforementioned patients not only suffer from severe malocclusion, wherein orthodontic treatment is rather limited, but also developing patients suffer from insufficient oxygenation. This occurrence is very severe, and some muscular disorders, such as bruxism and stomatognathic movement disorders, might follow. Therefore, a surgical approach is the preferable method of treatment in these patients [36,49]. Although in general bimaxillary procedures aim to improve bite and facial features, genioplasty advancing the tongue muscle could be an alternative in less severe cases. Improvements in airway function as well as facial features have been shown with genioplasty [46]. This protocol could be an alternative method as a less invasive approach to treatment and could be an interesting solution for individuals with TCS.

Proper sleep hygiene, including a minimum of 7 h of sleep per day, is crucial for general health. Good sleep quality regulates energy and memory. Furthermore, general health is influenced by sleep, specifically regarding the endocrine and immune systems. This is crucial for growing patients [1,50,51]. Hence, ensuring easier breathing and malocclusion corrections should be carried out in individuals in TCS with all due haste. Therefore, surgical methods of malocclusion treatment must be considered. This is all the more relevant in TCS patients, where, if it is applied, surgery yields an improvement in breathing during sleep. The current findings suggest that early airway intervention is crucial in the treatment of OSA among patients with TCS. In younger patients, usually, distraction osteogenesis is performed, whilst in adolescents, BSSO or bimaxillary surgery is taken into consideration [34,35]. Alternative upcoming treatment methods, such as HNS, could also be a promising solution in the near future. However, these above-mentioned methods are novel and do need further investigation. Additionally, certain pharmacological treatments, specifically those targeting the neuromuscular tonus of the airways, may be beneficial. Some patients may require a combination of methods, especially if their breathing problem is severe. The method of choice for treatment is tracheotomy in this case, but some individuals may refuse it. For this reason, combination therapy of bilateral HGNS and CPAP could be an alternative method of treatment [45].

Patients with congenital diseases are usually treated by a specialised team [8]. Furthermore, TCS is uncommon, with its prevalence being estimated at ca. 1:50,000 live births. Due to the fact that this congenital syndrome is a rare condition, there is no one specific and uniform treatment plan. The observed infrequency of data on this topic was therefore the biggest limitation of the present study. The authors, via the collection and presentation of possible methods of treatment, aimed to illustrate the numerous approaches to its clinical management, with the hope that the difficulties which TCS patients with OSA face will be resolved or lessened. In order to make this a more tangible goal, long-term follow-up studies on the outcomes of MDO should be executed. Finally, customised CPAP interfaces, directed towards rarer conditions such as craniofacial anomalies, are also expected to undergo great development in the future.

## 5. Conclusions

Obstructive sleep apnoea is highly prevalent in Treacher Collins Syndrome, with the majority of issues correlating with the craniofacial abnormalities characteristic of the syndrome itself. Early diagnosis of OSA and expedited interventions are essential to preventing further complications, specifically systemic ones. Treatment of obstructive sleep apnoea is necessary even in young patients, with CPAP being a good first-choice alternative. Due to the increasing severity of facial malformations that comes with age, surgical methods are still the most effective for the treatment of OSA in TCS patients, with MDO being the mainstay. Emerging therapies such as hypoglossal nerve stimulation could prove to be beneficial in the future; however, the need for further study and investigation is, at present, necessary. Owing to the severity of its symptoms, a multidisciplinary approach is mandatory to improve patients’ outcomes. It is of paramount importance for its treatment to be highly individualised and to focus on not only improving its features but also increasing one’s quality of life.

## Figures and Tables

**Figure 1 jcm-14-04741-f001:**
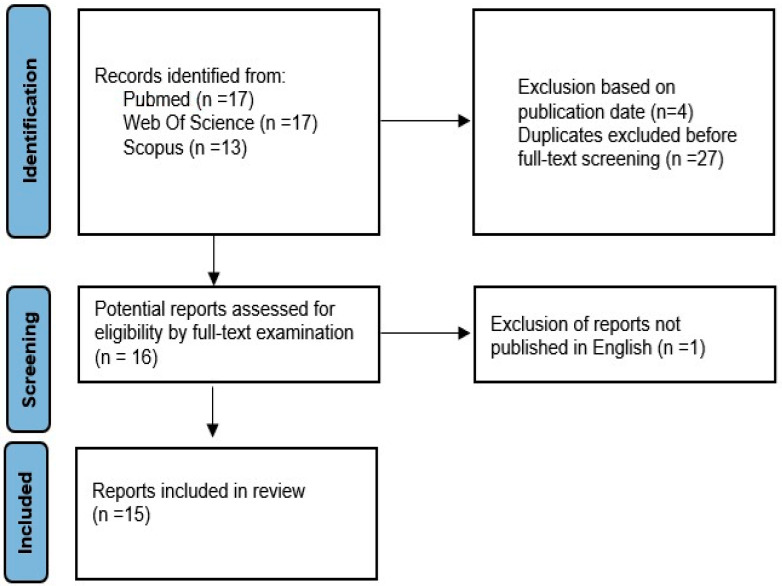
The PRISMA flow diagram illustrating the search criteria.

**Figure 2 jcm-14-04741-f002:**
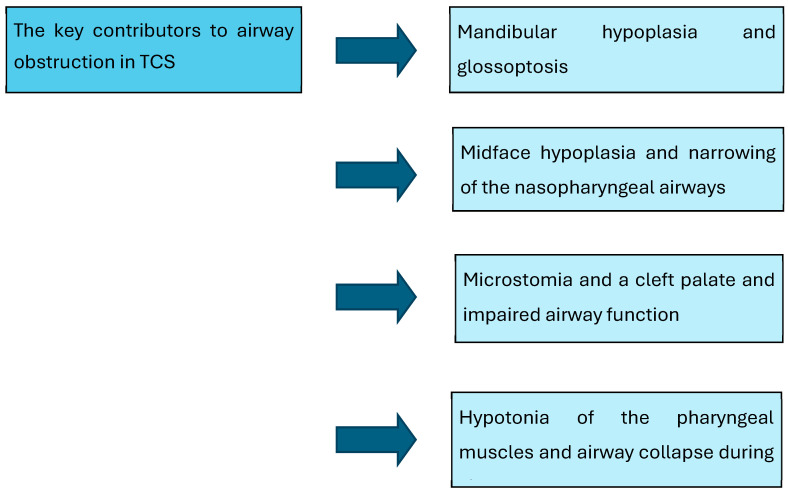
The key contributors to airway obstruction in TCS.

**Table 1 jcm-14-04741-t001:** Search strategy for OSA and TCS.

1.	“Treacher Collins Syndrome” AND “obstructive sleep apnoea”
2.	“Treacher Collins Syndrome” AND “OSA”
3.	“Craniofacial anomalies” AND “sleep-disordered breathing”
4.	“Mandibular distraction osteogenesis” OR “airway surgery”

**Table 2 jcm-14-04741-t002:** Non-surgical and surgical approaches to the treatment of OSA in patients with Treacher Collins Syndrome.

Non-surgical approaches	Continuous positive airway pressure (CPAP)
	Nasal airway stents
	Mandibular advancement devices (MADs)
Surgical interventions	Mandibular distraction osteogenesis (MDO)
	Lengthening the mandible to advance the tongue base and open the airways
	Tracheotomy
	Hypoglossal nerve stimulation (HNS)

## Data Availability

The data supporting the findings of this study are available from the corresponding author upon reasonable request.

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
