# Peer review of "Obstructive Sleep Apnoea in Patients with Treacher Collins Syndrome—A Narrative Review"

_jcm, 2025, doi:10.3390/jcm14134741_

Round 1
Reviewer 1 Report
Comments and Suggestions for Authors
- A brief summary - The primary aim of this semi-systematic review is to understand the prevalence, diagnosis and various treatment methods including advancements in patients with Treacher-Collins Syndrome (TCS) suffering with Obstructive sleep apnoea (OSA)
The main strength of this paper lies in its comprehensiveness owing to the limited research done previously.
- General concept comments
Review:
- TCS and OSA have been extensively researched individually, but the literature connecting them i.e. patients with TCS suffering with OSA is scarce. This aspect renders relevance to the review and bridges the gap in the knowledge that has been identified.
- A totality of 15 articles being selected shows the scarcity or lack of sufficient literature which necessitates further research. Even though similar research was published in the recent years, this review still maintains relevance due to constant evolution
- Majority of the references cited are from the past 6 years with the earliest being 2012, without excessive self-citations
- The tables/figures are all appropriate and self-explanatory.
Specific comments – However some of the undermentioned points can be considered to be a valuable addition to the manuscript
- The authors mentioned lack of patient compliance as the main disadvantage in using CPAP in Lines 178-179. The reason for the lack of compliance should be included as stated below.
“CPAP in general has been a great treatment methodology for OSA. Treating OSA in patients with Treacher Collins syndrome (TCS) remains challenging due to anatomical abnormalities. Many patients report intolerance to continuous positive airway pressure (CPAP) therapy, often due to mandibular hypoplasia, which contributes to poor mask fit and discomfort”
Miller SD , Glynn SF , Kiely JL , McNicholas WT . The role of nasal CPAP in obstructive sleep apnoea syndrome due to mandibular hypoplasia . Respirology. 2010. ; 15 ( 2 ): 377 – 379
- The limitation of Maxillo-Mandibular advancement surgery can be included in the manuscript.
Maxillo mandibular advancement surgery specifically in patients with Velo-paltal insufficiency may exacerbate the condition further worsening the insufficiency.
Wong ACL, Jones A, Stone A, MacKay SG. Combination CPAP and bilateral hypoglossal nerve stimulation for obstructive sleep apnea in Treacher Collins syndrome: first case report. J Clin Sleep Med. 2023 Jan 1;19(1):197-199.
- The novel hypoglossal nerve stimulation method has been studied extensively since 2020. The authors could include more advancements in this topic. I will add some important information that can be added with citations.
Hypoglossal nerve stimulation (HGNS) is generally categorized into two main types based on how the hypoglossal nerve is stimulated. Unilateral and bilateral.
Bilateral HGNS is the more advanced of the two types and the reason is as mentioned underneath. Even though all this information may not be included in the manuscript, the authors can benefit from the knowledge, mention the types and advantages of Bilateral HGNS
Complete concentric collapse (CCC) of the velopharyngeal airway is a common reason patients are excluded from receiving unilateral (HGNS) for obstructive sleep apnea. This is because unilateral HGNS only stimulates one side of the tongue, helping to open the airway at the tongue base, but it doesn’t effectively treat collapse that involves the soft palate and side walls of the throat, as seen in CCC. As a result, patients with CCC tend to respond poorly to this treatment. Identifying CCC using drug-induced sleep endoscopy (DISE) helps avoid implanting devices in patients unlikely to benefit. In contrast, bilateral HGNS which is a newer technique that stimulates both sides of the tongue, may better address CCC by creating more balanced and effective airway opening.
Vanderveken OM, et al. Clinical Predictors of the Success of Hypoglossal Nerve Stimulation for Obstructive Sleep Apnea. Otolaryngol Clin North Am. 2020;53(4):521–530.
Chiesa Estomba CM, et al. Genio™ System: The First European Experience with a Novel Bimodal Bilateral Hypoglossal Nerve Stimulation for OSA. Sleep Breath. 2021;25:1317–1323.
Lewis R , Le J , Czank C , Raux G . Control of OSA in a patient with CCC of soft palate using bilateral hypoglossal nerve stimulation . Clin Case Rep. 2021. ; 9 ( 4 ): 2222 – 2224 .
- The authors could also mention a combination therapy of Bilateral HGNS and CPAP together for patients with craniofacial syndromes who are intolerant of conventional therapies and decline tracheostomy. Including this in the discussion could broaden the treatment aspect of the OSA, particularly given the anatomical challenges in this patient population with TCS.
“A Recent case report has described clinical benefits of this approach, which includes significant reduction in snoring, improved sleep quality, and improved CPAP mask tolerance due to the submental activation patch used in Bilateral HGNS systems.
Wong ACL, Jones A, Stone A, MacKay SG. Combination CPAP and bilateral hypoglossal nerve stimulation for obstructive sleep apnea in Treacher Collins syndrome: first case report. J Clin Sleep Med. 2023 Jan 1;19(1):197-199.
- “Genioplasty”
It may be helpful for the authors to include some discussion on the challenges of managing recurrent sleep apnea in patients with craniofacial syndromes who have already undergone multiple facial bone surgeries. While initial improvements are often seen after maxillofacial advancement procedures, studies have shown that sleep apnea can return over time (Patel & Fearon, 2014). Since these patients frequently complete orthodontic treatment and undergo several major osteotomies at a young age, repeating such invasive procedures later on may carry risks, including changes to facial appearance or bite alignment. In these situations, genioplasty aimed at advancing the tongue muscle has been described as a less invasive option that can still provide meaningful improvement in airway function (Kino et al., 2023; Riley et al., 1983; Ueda et al., 1996). This approach does not significantly affect occlusion and tends to preserve facial aesthetics (Riley et al., 1987). The current study fits well within this context and could benefit from a brief mention of these alternative strategies.
Patel N, Fearon JA. Treatment of the syndromic midface: a long-term assessment at skeletal maturity. Plast Reconstr Surg. 2014;135:731e–742e
Kino H, Ueda K, Hirota Y, Okamoto T. Useful Genioplasty for Repeated Recurrent Sleep Apnea of Congenital Anomalies and Its Evaluation. Plast Reconstr Surg Glob Open. 2023 Mar 14;11(3)
Riley R, Guilleminault C, Powell NS, et al. Mandibular osteotomy and hyoid bone advancement for obstructive sleep Apnea. Sleep. 1983;7:79–82.
Ueda K, Tajima S, Tanaka Y, et al. Correction of severe sleep apnea in a case of Treacher-Collins syndrome. Eur J Plast Surg. 1996;19:320–322.
Riley RW, Powell N, Guilleminault C. Current surgical concepts for treating obstructive sleep apnea syndrome. J Oral Maxillofac Surg. 1987;45:149–157
- Sentence in lines 39-40 could be reconstructed to bring more clarity
“The medical field recognises the increasing prevalence of obstructive sleep apnoea (OSA) with the disorder involving excessive muscle relaxation during sleep.”
- Figure 2. The second box from the top in the key contributions to airway obstruction in TCS is Incomplete and needs to be corrected.
- Line 228. Is the term OXIDATION or OXYGENATION? Please clarify.
Author Response
Dear Reviewer,
please find the attached file as response to your review. Thank you once again for the time and effort.
Best regards. - Anna Paradowska-Stolarz

Reviewer 2 Report
Comments and Suggestions for Authors
While the manuscript provides a clinically important overview, I have identified several methodological and structural aspects that merit revision to ensure consistency with established standards for review articles.
Clarification of Review Type: The manuscript is currently presented as a "semi-systematic review." However, it does not fully adhere to the criteria of a systematic review or a scoping review. I think it's more a narrative review. The search strategy lacks transparency and reproducibility, the search terms were minimal and not database-specific, it is not comprehensive. Study selection and data extraction methods are insufficiently detailed. No formal quality appraisal or risk of bias assessment was performed. The authors should either:
-
Enhance methodological rigor and reframe the manuscript as a scoping review, aligning with PRISMA-ScR and JBI guidelines, or
-
Reclassify the manuscript as a narrative review with a kind of structured synthesis, which may be more appropriate given the exploratory scope and limited number of included studies.
In its current state it is not possible to conduct a more detailed review of your article. Consider to clarify its objective.
Author Response
Dear Reviewer,
thank you for an effort given into suggestions to our article. Please, find attached file with comments to that. We are waiting for further suggestions.
With best regards - Anna Paradowska-Stolarz

Round 2
Reviewer 1 Report
Comments and Suggestions for Authors
Dear Authors,
Thank you for taking into consideration my recommendations and modifying the manuscript.
Reviewer 2 Report
Comments and Suggestions for Authors
Dear authors,
Thank you for the efforts on improving the manuscript. I have no further inquiries regarding your work. Well done! However, I suggest you to improve a little bit the quality of the english language using professional editing services, althoug there is no mandatory.